

# A structure-preserving linearly homomorphic signature scheme with designated combiner

Xuan Zhou[1],[*], Yuan Tian[1],[*], Weidong Zhong[1], Tanping Zhou[2] and Xiaoyuan Yang[1]

[1] College of Cryptography Engineering, Engineering University of People's Armed Police, Xi'an, Shanxi, China
[2] TCA Laboratory, State Key Laboratory of Computer Science, Institute of Software, Chinese Academy of Sciences, Beijing, China
[*] These authors contributed equally to this work.

## ABSTRACT

Linearly homomorphic signature (LHS) allows the acquisition of a new legal signature using the homomorphic operation of the original signatures. However, the public composability of LHS also prevents it from being used in some scenarios where the combiner needs to be designated. The LZZ22 scheme designates a combiner and preserves the signature structure by having the signer and the designated combiner share a secret. However, LZZ22 is not secure enough because the secret is constant. Here, we first prove that there is a polynomial time adversary that can crack the secret in LZZ22 through multiple signature queries. Then, we propose a new scheme, which realizes all the functions of LZZ22 and fixes the security problem by changing the secret with the message. The proposed scheme is shown to be secure against existential forgery on adaptively chosen subspace attacks under the random oracle model. Finally, we detail how to apply our scheme to the proxy signature and perform it on a personal computer, and the results show that our scheme is efficient.

# INTRODUCTION

Linear network coding is an effective technique to improve network throughput. It allows nodes to combine multiple received data packets into one packet and forward it, so as to realize efficient data transmission. However, some malicious nodes in the network may inject forged packets into legitimate packets, and synthesize a corrupted packet that can be forwarded to other nodes. Other nodes in the network combine the corrupted packet with the legitimate packets to synthesize a new corrupted packet and forward it. Due to the nature of network coding, corrupted packets will pollute more legitimate packets, leaving the destination node unable to recover the original data. This type of attack is called a pollution attack. The digital signature (*Diffie & Hellman, 1976*) is one of the core technologies of cryptography, which can provide authenticity, integrity, and non-repudiation of information. However, the general digital signature scheme cannot be used

Corresponding authors
Weidong Zhong, wdeast@163.com
Tanping Zhou, tanping2020@iscas.ac.cn

to solve the pollution attack problem because the original signature becomes invalid once the message is changed. The homomorphic signature (HS) is a type of digital signature that allows any entity to obtain a new legal signature by homomorphic operation on the original signature. Among them, the linearly homomorphic signature (LHS) (*Attrapadung, Libert & Peters, 2013*) can well resist pollution attacks in network coding because it supports linear homomorphic operations on messages (*Zhao et al., 2007*; *Charles, Jain & Lauter, 2006*; *Yu et al., 2008*; *Yun, Cheon & Kim, 2010*). With the development of homomorphic signature technology, LHS has also been used in scenarios such as electronic health systems (*Li, Zhang & Sun, 2021*), blockchain (*Lin et al., 2018*), and the Internet of Things (IoT) (*Li, Zhang & Liu, 2020*).

According to the homomorphism of LHS, any entity can obtain the signature of the linear combination of the original messages using the homomorphism operation of the obtained signatures from a set of message/signature pairs with the same label. This is the public composability of LHS. However, in some scenarios such as proxy signing, the user will designate a server that has the unique authority to combine messages and generate a legitimate signature. This allows the designated server to sign instead of the user in special circumstances, such as when it is not convenient for the signer, or if there is too much data. General LHS cannot implement the function of designating a combiner due to its public composability. Designating a combiner means that the signature is homomorphic for the combiner but not for other entities in the system. The linearly homomorphic signature with designated combiner (LHSDC) (*Lin, Xue & Huang, 2021*) realizes the function of designating a combiner by key agreement. However, the signature structure generated by the combiner was changed (*Lin, Xue & Huang, 2021*), so that the combined signature cannot continue to be used as the input of the combination algorithm. *Li, Zhang & Zhang (2022)* proposed the formal definition and security model of structure-preserving linearly homomorphic signature scheme with a designated combiner (SPS-LHSDC) and constructed the first SPS-LHSDC scheme, LZZ22. LZZ22 modifies the signature algorithm based on *Lin, Xue & Huang (2021)* to make the combined signature structure consistent with the original signature structure. However, this scheme has a security problem.

## Our contributions

In this article, we first prove that there is a polynomial time adversary that can crack the secret information in LZZ22 through multiple signature queries. Then, the adversary is able to forge the signature corresponding to any message.

Secondly, we propose a new scheme, which has all the functions of LZZ22 and fixes the security problem by changing the secret information with the message by adding one hash operation and one exponential operation to the signature algorithm. Meanwhile, we detail how to apply our scheme to the proxy signature.

Finally, we run the signature forgery program of LZZ22 through experiments, and the results show that the time required to forge a signature is inversely proportional to the message dimension. We run the proposed scheme in the same experimental environment

and compare it with other LHS schemes. The experimental results show that the signature algorithm and the verification algorithm of our scheme are efficient, and that the usage of system resources by our algorithm is low.

## Related works

*Desmedt (1993)* introduced the concept of HS, and *Johnson et al. (2002)* introduced its formal definition and general framework in 2002. Afterward, many HS schemes appeared (*Cheng et al., 2016*; *Li et al., 2018*; *SadrHaghighi & Khorsandi, 2016*; *Catalano, Fiore & Warinschi, 2014*; *Gorbunov, Vaikuntanathan & Wichs, 2015*; *Zhang, Jianping & Ting, 2012*; *Aranha & Pagnin, 2019*), including one in which, LHS supports linear homomorphic operations on messages. However, the early LHS schemes lack strict security proof and are not practical. In 2009, *Boneh et al. (2009)* constructed the first provably secure LHS scheme under the random oracle model. In this scheme, each file is regarded as a linear vector subspace, and the source node signing the basis vectors of the subspace is equivalent to signing the whole file. *Gennaro et al. (2010)* proposed the first LHS scheme based on the RSA difficult problem. This scheme reduces the cost compared with the scheme in *Boneh et al. (2009)*. To achieve the function of anti-quantum attacks, *Boneh & Freeman (2011)* proposed the first lattice-based LHS scheme in 2011. The security of the scheme is based on the SIS difficulty problem. The signature verification of the scheme is completed in the binary domain. *Chen, Lei & Qi (2016)* constructed the first LHS scheme based on the SIS difficulty problem under the standard model. This scheme can resist weak adversaries and provide weak context-hidden privacy. In 2018, *Lin et al. (2018)* constructed the first ID-based LHS scheme by introducing ID-based signature technology. This scheme uses the user's identity ID as the public key, which avoids the disadvantage of difficult key management. *Zhang et al. (2018)* proposed a more efficient ID-based LHS scheme, Zhang18. However, their scheme does not augment the original vector, so it is not suitable for network coding. Moreover, ID-based LHS schemes suffer from key escrow problems. In 2021, *Wu, Wang & Yao (2021)* constructed a certificate-free LHS scheme Wu21 for network coding. Their scheme avoids both the certificate management problem and the key escrow problem.

*Lin, Xue & Huang (2021)* and *Lin et al. (2017)* proposed two LHSDC schemes to make the LHS scheme applicable in scenarios that need to designate a combiner or verifier, Lin17 and Lin21. The latter made up for the former's lack of public verifiability. In both schemes, only the designated combiner has the right to combine the original signature. However, the combined signature structure in *Lin, Xue & Huang (2021)* and *Lin et al. (2017)* has changed, so that the combined signature can no longer be used as the input of the combination algorithm. In 2022, *Li, Zhang & Zhang (2022)* proposed the first SPS-LHSDC scheme LZZ22 based on the Lin21. In the SPS-LHSDC scheme, the combined signature has the same signature structure as the original signature, so it can still be used as the input of the combination algorithm. This function enables the SPS-LHSDC scheme to be used in certain scenarios. However, the scheme LZZ22 has a security problem.

## Organization

The overall structure of the rest of this article is as follows. In the "Preliminaries" section, we introduce some preliminaries. In the "The Security Problem of LZZ22", we analyze the security of LZZ22 and then construct a signature forgery algorithm for LZZ22. In "The proposed Scheme" section, we propose a new SPS-LHSDC scheme and prove the correctness and security of the scheme. In the "Application and Security Analysis" section, we first describe how to apply the scheme in this article to proxy signature, and then run the signature forgery experiment of LZZ22 and compared the efficiency of our scheme with four other LHS schemes. Finally, we summarize the full text and describe future research directions in the Conclusions.

## PRELIMINARIES

Here, we introduce some basics, including symmetric bilinear mapping, the augmented basis vector, and the formal definition of SPS-LHSDC.

### Symmetric bilinear mapping

In 1991 *Menezes, Vanstone & Okamoto (1991)* proposed symmetric bilinear mapping which is defined as follows.

Let $\mathbb{G}_1$ and $\mathbb{G}_2$ be groups of order $q$. If a mapping $e : \mathbb{G}_1 \times \mathbb{G}_1 \to \mathbb{G}_2$ satisfies:

1. Calculability: $\forall g \in \mathbb{G}_1$, solving $e(g, g)$ is efficient;
2. Bilinear: $\forall a, b \in \mathbb{Z}_q$, $g \in \mathbb{G}_1$, all satisfy $e(g^a, g^b) = e(g, g)^{ab}$;
3. Non-degenerate: $\exists g \in \mathbb{G}_1$, makes $e(g, g) \neq 1$.

The mapping is called symmetric bilinear mapping.

***Definition 1*: (computational Diffie-Hellman problem (CDH))** (*Boneh, 1998*). Given a triple $(g, g^a, g^b)$, where $g$ is the generator of $\mathbb{G}_1$, $a, b \leftarrow_R \mathbb{Z}_q^*$ are two unknown elements, solve $g^{ab}$.

***Definition 2*: (CDH assumption)** (*Boneh, 1998*). If for any probabilistic polynomial-time (PPT) algorithm $\mathcal{A}$, the probability of solving the CDH problem is negligible, then it is difficult to solve the CDH problem in $\mathbb{G}_1$.

### The augmented basis vector

In an LHS scheme, a file is usually divided into a set of $n$-dimensional original vectors $\overline{\mathbf{v}}_1, \overline{\mathbf{v}}_2, \ldots, \overline{\mathbf{v}}_m \in \mathbb{Z}_q^n$, where $\overline{\mathbf{v}}_i = (v_{i1}, v_{i2}, \ldots, v_{in})$, $i \in \{1, 2, \ldots, m\}$, $q$ is a large prime. To ensure that receivers in the network can recover this set of vectors, the set of original vectors will be augmented to ensure that they are linearly independent. The augmentation operation of this set of vectors is as follows (*Boneh et al., 2009*): For each $i \in \{1, 2, \ldots, m\}$, let

$$\mathbf{v}_i = \left(v_{i1}, \ldots, v_{in}, v_{i(n+1)}, \ldots, v_{i(n+m)}\right) \in \mathbb{Z}_q^N, \tag{1}$$

where

$$v_{i(n+j)} = \begin{cases} 1, & j = i \\ 0, & j \neq i \end{cases} \qquad i, j = 1, 2, \ldots, m. \tag{2}$$

Among them, $N = n + m$, add a $m$-dimensional unit vector (the $i$-th bit of this unit vector is "1", and the rest bits are "0") after the basis vector $\overline{\mathbf{v}}_i$. This set of vectors after the augmentation operation becomes a set of basis vectors of the subspace to which the original file belongs due to its linear-independent property.

**The formal definition of SPS-LHSDC**

*Definition 3*: The SPS-LHSDC scheme consists of five PPT algorithms (*Li, Zhang & Zhang, 2022*):

- **Setup**$(1^\lambda, N) \to (pp)$: The algorithm inputs the security parameter $1^\lambda$ and the dimension $N$, outputs the system public parameter $pp$;
- **KeyGen**$(pp) \to (sk, pk)$: The algorithm inputs $pp$ and outputs a private key $sk$ and the corresponding public key $pk$;
- **Sign**$(pp, sk_A, pk_B, id, \mathbf{v}_k) \to (\tau, \sigma_k)$: The algorithm inputs $pp$, the signer's private key $sk_A$, the combiner's public key $pk_B$, file identifier $id$ and vector $\mathbf{v}_k$, and outputs the subspace label $\tau$ and signature $\sigma_k$;
- **Combine**$(pp, pk_A, sk_B, \tau, \{(\mathbf{v}_k, \sigma_k, \beta_k)\}_{k=1}^m) \to (\mathbf{v}, \sigma)$: The algorithm inputs $pp, pk_A, sk_B$, $\tau$ and $m$ triples $\{(\mathbf{v}_k, \sigma_k, \beta_k)\}_{k=1}^m$, where $\beta_k \in \mathbb{Z}_q^*$, outputs a message/signature pair $(\mathbf{v}, \sigma)$;
- **Verify**$(pp, pk_A, \tau, \mathbf{v}, \sigma) \to (0, 1)$: The algorithm inputs $pp, pk_A, \tau, \mathbf{v}$, and $\sigma$. If $\sigma$ is the legal signature of vector $\mathbf{v}$, the algorithm outputs 1; otherwise, the algorithm outputs 0.

*Correctness*

The SPS-LHSDC scheme is correct if it satisfies the following two conditions:

1. $\forall id \in \{0, 1\}^\lambda$ and $\mathbf{v}_k \in \mathbb{Z}_q^N$, if $\sigma_k \leftarrow$ **Sign**$(pp, sk_A, pk_B, id, \mathbf{v}_k)$, then

$$\text{Verify}(pp, pk_A, \tau, \mathbf{v}_k, \sigma_k) = 1. \tag{3}$$

2. $\forall id \in \{0, 1\}^\lambda$ and $\{(\mathbf{v}_k, \sigma_k, \beta_k)\}_{k=1}^m$, if **Verify**$(pp, pk_A, \tau, \mathbf{v}_k, \sigma_k) = 1$ holds for all $k \in \{1, \ldots, m\}$, then

$$\text{Verify}(pp, pk_A, \tau, \text{Combine}(pk_A, sk_B, id, \{(\mathbf{v}_k, \sigma_k, \beta_k)\}_{k=1}^m) = 1. \tag{4}$$

*Security model*

In the SPS-LHSDC scheme, the forgery of adversary $\mathcal{A}$ is said to be successful if the forged message/signature pair can pass the verification algorithm, and the forgery conforms to one of the following types of forgery.

**Type 1 Forgery:** The adversary $\mathcal{A}$ never queried the subspace $V$ and generates a valid signature for $\mathbf{w}^* \in V$, where $\mathbf{w}^* \neq \mathbf{0}$.

**Type 2 Forgery:** The adversary $\mathcal{A}$ has queried the subspace $V$ labeled $\tau$, and then $\mathcal{A}$ uses the label $\tau$ to generate a valid signature for $\mathbf{w}^* \notin V$, where $\mathbf{w}^* \neq \mathbf{0}$.

**Type 3 Forgery:** The adversary $\mathcal{A}$ has queried the subspace $V$, and then $\mathcal{A}$ generates a valid signature for $\mathbf{w}^* \in V$ without knowing the private key of the combiner, where the vector $\mathbf{w}^*$ is composed of the basis vector of $V$ and $\mathbf{w}^* \neq \mathbf{0}$.

*Definition 4*: If the probability of any PPT adversary $\mathcal{A}$ winning the following games is negligible, then the SPS-LHSDC scheme is safe.

- **Setup:** The challenger $\mathcal{C}$ selects the security parameter $1^\lambda$ and a positive integer $N$ and runs **Setup**$(1^\lambda, N) \to (pp)$, **KeyGen**$(pp) \to (sk, pk)$ in turn. Then $\mathcal{C}$ sends $(pp, pk_A)$ to $\mathcal{A}$.

- **Combiner-key Generation Query:** when $\mathcal{A}$ initiates this query, $\mathcal{C}$ runs **KeyGen**$(pp) \to (sk_B, pk_B)$ and sends $pk_B$ to $\mathcal{A}$.

- **Combiner Corruption Query:** When $\mathcal{A}$ initiates this query, $\mathcal{A}$ sends a combiner's public key $pk_B$ to $\mathcal{C}$, then $\mathcal{C}$ returns the corresponding $sk_B$ to $\mathcal{A}$.

- **Sign Query:** When $\mathcal{A}$ initiates this query, $\mathcal{A}$ selects a subspace $\overline{V_i} = span\{\overline{\mathbf{v}}_{i1}, \dots, \overline{\mathbf{v}}_{im}\}$, where $\overline{\mathbf{v}}_{i1}, \dots, \overline{\mathbf{v}}_{im} \in \mathbb{Z}_q^n$, then $\mathcal{C}$:
  1) For each $k \in \{1, \dots, m\}$, augments $\overline{\mathbf{v}}_{ik} \in \mathbb{Z}_q^n$ to $\mathbf{v}_{ik} \in \mathbb{Z}_q^N$, then gets a new subspace $V_i$.
  2) Randomly selects the file identifier $id_i$, then gets the subspace label $\tau_i = (id_i, pk_B)$ for $V_i$.
  3) For each $k \in \{1, \dots, m\}$, runs **Sign**$(pp, sk_A, pk_B, id_i, \mathbf{v}_{ik}) \to \sigma_{ik}$.
  4) Returns $\tau_i$ and $\sigma_i = (\sigma_{i1}, \dots, \sigma_{im})$ to $\mathcal{A}$.

- **Combine Query:** When $\mathcal{A}$ initiates this query, $\mathcal{A}$ sends $\left(\tau_i, \{(\mathbf{v}_{ik}, \sigma_{ik}, \beta_{ik})\}_{k=1}^m\right)$ to $\mathcal{C}$, then $\mathcal{C}$ runs **Combine**$\left(pp, pk_A, sk_B, \tau_i, \{(\mathbf{v}_{ik}, \sigma_{ik}, \beta_{ik})\}_{k=1}^m\right) \to (\mathbf{v}, \sigma)$ and returns $(\mathbf{v}, \sigma)$ to $\mathcal{A}$.

- **Forgery:** $\mathcal{A}$ outputs a signer's public key $pk_A^*$, a combiner's public key $pk_B^*$, a subspace label $\tau^*$, a non-zero vector $\mathbf{v}^* \in \mathbb{Z}_q^N$ and a signature $\sigma^*$.

If **Verify**$(pp, pk_A^*, \tau^*, \mathbf{v}^*, \sigma^*) = 1$ and one of the following three conditions is true, the adversary $\mathcal{A}$ is considered to win the above game:

1. $\tau^* \neq \tau_i$ for any $i$;

2. $\tau^* = \tau_i$ for some $i$ but $\mathbf{v}^* \notin V_i$;

3. $\tau^* = \tau_i$ for some $i$, $\mathbf{v}^* \in V_i - \{\mathbf{v}_{i1}, \dots, \mathbf{v}_{im}\}$, and $\mathcal{A}$ has not queried the combiner's private key.

# THE SECURITY PROBLEM OF LZZ22

Here, we first analyze the security of LZZ22 and then perform signature forgery against its security problem.

## Security analysis of the LZZ22

Li et al. used a similar method as *Boneh et al. (2009)* to prove that LZZ22 is secure against existential forgery on adaptively chosen subspace attacks under the random oracle model. However, the security model of the SPS-LHSDC scheme has one more type of forgery (Type 3 Forgery) than *Boneh et al. (2009)*. *Li, Zhang & Zhang (2022)* actually only proves that the LZZ22 scheme can resist Type 1 Forgery and Type 2 Forgery. Type 3 Forgery exists

in SPS-LHSDC because requires that no entity other than the designated combiner combine original signatures to generate a new signature. Type 3 Forgery without knowing the private key of the designated combiner, can obtain the signature of a new message. Below, we analyze the feasibility of Type 3 Forgery in LZZ22.

LZZ22 designates a combiner by binding the combiner's public key to the signature algorithm. The signer then shares a secret information with the combiner through key negotiation. The secret information can make the signatures generated by the signer temporarily lose the homomorphic property. Therefore, other entities in the system cannot combine signatures, and the combiner who has the secret information can restore the homomorphic property of the signature, thereby obtaining the authority to combine signatures. Thus, the key to realizing the function of designating a combiner is that only the signer and the designated combiner have the secret information. An adversary that can decrypt the secret information in LZZ22 can pretend to be the designated combiner and make any combination of the original signatures to forge a new message/signature pair. Next, we explore a theoretical way to crack LZZ22's secret information.

In LZZ22, the document will be divided into $m$ message vectors $\{\mathbf{v}_i\}_{i=1}^m$, where $\mathbf{v}_i = (v_{i1}, v_{i2}, \ldots, v_{i,m+n})$, $i \in \{1, 2, \ldots, m\}$, and the signer signs these message vectors, respectively. The signature corresponding to the $k$-th message vector is $\sigma_k = \left( \prod_{i=1}^n g_i^{v_{k,i}} \prod_{j=1}^m H_1(\tau, j)^{v_{k,n+j}} u_B \right)^{a_A}$, where $\{g_i\}_{i=1}^n$ are the generators of $\mathbb{G}_1$, $\tau$ is the subspace label, $H_1$ is a hash map: $\{0, 1\}^* \to \mathbb{G}_1$, $u_B$ is the public key of the designated combiner, $a_A$ is the private key of the signer. $u_B{}^{a_A} = u_A{}^{a_B} = g^{a_A a_B}$, so $u_B{}^{a_A}$ is the secret information shared by the signer and the designated combiner. However, $u_B{}^{a_A}$ is a fixed value and can be obtained by division between two different signatures. If the adversary finds out the value of $u_B{}^{a_A}$, he obtains the authority to combine signatures, thereby forging a new message/signature pair.

## Signature forgery

The specific requirements of Type 3 Forgery in the SPS-LHSDC security model (*Li, Zhang & Zhang, 2022*) are as follows:

- The adversary $\mathcal{A}$ does not know the private key of the combiner.
- $\mathcal{A}$ has queried the subspace $V$, that is, $\mathcal{A}$ knows a set of basis vectors $\{\mathbf{v}_i\}_{i=1}^m$ of subspace $V$ and the corresponding signature $\{\sigma_i\}_{i=1}^m$.
- $\mathcal{A}$ generates a legal signature for a non-zero vector $\mathbf{v}^*$, and $\mathbf{v}^*$ must be obtained by a linear combination of $\{\mathbf{v}_i\}_{i=1}^m$.

According to the requirements of Type 3 Forgery and the security vulnerability of the LZZ22 scheme, if the adversary $\mathcal{A}$ wants to forge the signature $\sigma^*$ of a vector $\mathbf{v}^*$, $\mathcal{A}$ first needs to find the secret information $u_B^{a_A}$ shared by the signer and combiner by asking for two different signature values (**Step 1**). The adversary then represents the attempted forged vector $\mathbf{v}^*$ with a set of basis vectors $\{\mathbf{v}_i\}_{i=1}^m$ of subspace $V$ (**Step 2**). Finally, the adversary,

with the secret information $u_B^{a_A}$, will be able to assume the identity of the combiner and run the Combine in LZZ22 to obtain the legal signature of the vector $\mathbf{v}^*$ (**Step 3**). The specific steps are as follows:

**Step 1:** Queries the signature $\sigma_0$ of any message $\mathbf{v}_0$ and the signature $\sigma_{00} = \frac{\sigma_0^2}{u_B^{a_A}}$ of message $2\mathbf{v}_0$; get $u_B^{a_A} = \frac{\sigma_0^2}{\sigma_{00}}$.

**Step 2:** After querying the subspace $V$ where the message $\mathbf{v}^*$ is located, a set of basis vector/signature pairs $\{\mathbf{v}_i, \sigma_i\}_{i=1}^m$ of the subspace $V$ is obtained, and $\mathbf{v}^*$ can be decomposed into

$$\mathbf{v}^* = \sum_{i=1}^m \beta_i \mathbf{v}_i. \tag{5}$$

**Step 3:** Calculates $\sigma_i^* = \frac{\sigma_i}{u_B^{a_A}}, i = 1, 2, \ldots, m$, respectively, and gets the signature corresponding to $\mathbf{v}^*$:

$$\begin{aligned}
\sigma^* &= \left(\prod_{i=1}^m \sigma_i^{*\beta_i}\right) \cdot u_B^{a_A} \\
&= \left(\prod_{i=1}^m \left(\frac{\sigma_i \sigma_{00}}{\sigma_0^2}\right)^{\beta_i}\right) \cdot \frac{\sigma_0^2}{\sigma_{00}}.
\end{aligned} \tag{6}$$

The correctness of the message/signature pair $(\mathbf{v}^*, \sigma^*)$ is obvious. The key to the successful forgery above is that the adversary $\mathcal{A}$ finds out the secret information $u_B^{a_A}$ shared by the signer and the designated combiner. This type of forgery satisfies the condition of the Type 3 forgery.

In this section, we find that the root cause of the insecurity of the LZZ22 scheme is that the secret information $u_B{}^{a_A}$ shared by the signer and the specified combinator is a fixed value, and this fixed value can be easily separated from the signature. Our signature forgery approach can attack some digital signature schemes with the same characteristics: (1) some of the important information in the signature is a fixed value; (2) this fixed value can be derived by arithmetic among multiple signatures. In the next section, we propose a more secure scheme. Compared with the LZZ22 scheme, our scheme adds a hash function value of the message $\mathbf{v}_k$ to the index part of the secret information $u_B{}^{a_A}$. If the adversary wants to obtain the secret information, it will need to solve the discrete logarithm problem, so our scheme ensures the security of secret information.

## THE PROPOSED SCHEME

Here, we first propose a new scheme by fixing the security problem of LZZ22, then we prove the correctness and security of our scheme.

### Construction

The proposed scheme is composed of five algorithms, Setup is responsible for generating initialization parameters, KeyGen is responsible for generating public/private keys of the user and the designated combiner, Sign is run by the signer and is responsible for

generating the original signature, Combine is run by the designated combiner and is responsible for generating the combined signature from the original signature. The Verify algorithm is responsible for verifying the legitimacy of all signatures. The details of each algorithm are as follows:

- **Setup**$(1^\lambda, N) \to (pp)$: The algorithm inputs the security parameter $1^\lambda$ and a positive integer $N$, then:
  1) Two multiplicative cyclic groups $\mathbb{G}_1$ and $\mathbb{G}_2$ with large prime $q$ are randomly selected, where $q > 2^\lambda$, a bilinear mapping $e : \mathbb{G}_1 \times \mathbb{G}_1 \to \mathbb{G}_2$.
  2) The generators $g, g_1, \ldots g_N$ is randomly selected in the group $\mathbb{G}_1$.
  3) Selects two hash function $H_1 : \{0,1\}^* \to \mathbb{G}_1$ and $H_2 : \mathbb{Z}_q^N \to \mathbb{Z}_q^*$.
  4) Outputs system public parameter $pp = (\mathbb{G}_1, \mathbb{G}_2, q, e, g, g_1, \ldots g_N, H_1, H_2)$.

- **KeyGen**$(pp) \to (sk, pk)$: The algorithm inputs $pp$, when the signer runs the algorithm, randomly selects $\alpha_A \in \mathbb{Z}_q^*$ as the signer's private key $sk_A$, and calculates $u_A = g^{\alpha_A}$ as the signer's public key $pk_A$; when the designated combiner runs the algorithm, randomly selects $\alpha_B \in \mathbb{Z}_q^*$ as the designated combiner's private key $sk_B$, calculates $u_B = g^{\alpha_B}$ as the public key $pk_B$ of the designated combiner.

- **Sign**$(pp, sk_A, pk_B, id, \mathbf{v}_k) \to (\tau, \sigma_k)$: The algorithm inputs $pp$, $sk_A = a_A$, $pk_B = u_B$, the file identifier $id \in \{0,1\}^\lambda$ and the vector $\mathbf{v}_k \in \mathbb{Z}_q^N$, then outputs the subspace label $\tau = (id, pk_B)$ and the signature

$$
\sigma_k = \left( \prod_{i=1}^{n} g_i^{v_{k,i}} \prod_{j=1}^{m} H_1(\tau, j)^{v_{k,n+j}} u_B^{H_2(\mathbf{v}_k)} \right)^{a_A}. \tag{7}
$$

- **Combine**$\left(pp, pk_A, sk_B, \tau, \{(\mathbf{v}_k, \sigma_k, \beta_k)\}_{k=1}^{m}\right) \to (\mathbf{v}, \sigma)$: The algorithm inputs $pp$, $pk_A = u_A$, $sk_B = a_B$, $\tau$, and $m$ triples $\{(\mathbf{v}_k, \sigma_k, \beta_k)\}_{k=1}^{m}$, where $\beta_k \in \mathbb{Z}_q^*$. The designated combiner calculates and outputs:

$$
\mathbf{v} = \sum_{k=1}^{m} \beta_k \mathbf{v}_k, \tag{8}
$$

$$
\sigma = u_A^{a_B H_2(\mathbf{v})} \prod_{k=1}^{m} \left( \sigma_k (u_A^{a_B H_2(\mathbf{v}_k)})^{-1} \right)^{\beta_k}. \tag{9}
$$

- **Verify**$(pp, pk_A, \tau, \mathbf{v}, \sigma) \to (0, 1)$: The algorithm inputs $pp$, $pk_A = u_A$, $\tau$, the vector $\mathbf{v}$, and the signature $\sigma$. If $e(\sigma, g) = e\left( \prod_{i=1}^{n} g_i^{v_i} \prod_{j=1}^{m} H_1(\tau, j)^{v_{n+j}} u_B^{H_2(\mathbf{v})}, u_A \right)$ holds, the algorithm outputs 1; otherwise, it outputs 0.

## Correctness

The correctness of the proposed scheme consists of two parts, namely the correctness of the signature algorithm and the correctness of the combination algorithm.

### The correctness of the signature algorithm

$\forall id \in \{0,1\}^\lambda$ and $\mathbf{v}_k \in \mathbb{Z}_q^N$, if $\sigma_k \leftarrow \mathbf{Sign}(pp, sk_A, pk_B, id, \mathbf{v}_k)$ holds, then

$$
\begin{aligned}
\gamma_1 &= e(\sigma_k, g) \\
&= e\left(\left(\prod_{i=1}^n g_i^{v_{k,i}} \prod_{j=1}^m H_1(\tau,j)^{v_{k,n+j}} u_B^{H_2(\mathbf{v}_k)}\right)^{a_A}, g\right) \\
&= e\left(\prod_{i=1}^n g_i^{v_{k,i}} \prod_{j=1}^m H_1(\tau,j)^{v_{k,n+j}} u_B^{H_2(\mathbf{v}_k)}, u_A\right) \\
&= \gamma_2.
\end{aligned}
\tag{10}
$$

### The correctness of the combination algorithm

$\forall id \in \{0,1\}^\lambda$ and $\{(\mathbf{v}_k, \sigma_k, \beta_k)\}_{k=1}^m$, if $\mathbf{Verify}(pp, pk_A, \tau, \mathbf{v}_k, \sigma_k) = 1$ holds for all $k \in \{1, \ldots, m\}$, $(\mathbf{v}, \sigma) \leftarrow \mathbf{Combine}(pk_A, sk_B, id, \{(\mathbf{v}_k, \sigma_k, \beta_k)\}_{k=1}^m)$, where $\mathbf{v} = \sum_{k=1}^m \beta_k \mathbf{v}_k$, then

$$
\begin{aligned}
\sigma &= \left(\prod_{i=1}^n g_i^{v_i} \prod_{j=1}^m H_1(\tau,j)^{v_{n+j}} u_B^{H_2(\mathbf{v})}\right)^{a_A} \\
&= u_B^{a_A H_2(\mathbf{v})} \left(\prod_{i=1}^n g_i^{v_i} \prod_{j=1}^m H_1(\tau,j)^{v_{n+j}}\right)^{a_A} \\
&= u_A^{a_B H_2(\mathbf{v})} \left(\prod_{i=1}^n g_i^{\prod_{k=1}^m \beta_k v_{k,i}} \prod_{j=1}^m H_1(\tau,j)^{\prod_{k=1}^m \beta_k v_{k,n+j}}\right)^{a_A} \\
&= u_A^{a_B H_2(\mathbf{v})} \sum_{k=1}^m \left(\left(\prod_{i=1}^n g_i^{v_{k,i}} \prod_{j=1}^m H_1(\tau,j)^{v_{k,n+j}}\right)^{a_A}\right)^{\beta_k} \\
&= u_A^{a_B H_2(\mathbf{v})} \sum_{k=1}^m \left(\sigma_k \cdot \left(u_A^{a_B H_2(\mathbf{v}_k)}\right)^{-1}\right)^{\beta_k}.
\end{aligned}
\tag{11}
$$

## Security analysis

In this section, we use a game to prove the security of the scheme. Our general idea is to assume that there exists a PPT adversary $\mathcal{A}$ that can forge a message/signature pair of our scheme with a non-negligible probability $\epsilon$. Then we will show that there exists another PPT algorithm $\mathcal{B}$, and that $\mathcal{B}$ can crack the CDH problem by interacting with $\mathcal{A}$ with another non-negligible probability $\epsilon'$. According to the CDH assumption that there exists no PPT algorithm that can crack the CDH problem with a non-negligible probability, therefore, we conclude that there exists no PPT adversary $\mathcal{A}$ that can achieve forgery with a non-negligible probability, thus proving the security of this scheme. In our proof process, we first define the type of queries that adversary $\mathcal{A}$ is able to make (capabilities of $\mathcal{A}$) and the way $\mathcal{B}$ replies, and find the probability $\epsilon_1$ that $\mathcal{B}$ is able to successfully simulate the system based on the way $\mathcal{B}$ replies (the probability that $\mathcal{B}$ has not given up the simulation).

Then, assuming that $\mathcal{A}$ has output a valid forgery with probability $\epsilon$, we find the probability $\epsilon_2$ that $\mathcal{B}$ correctly outputs a solution to the CDH problem using the forgery of $\mathcal{A}$. Finally, if all of the above events hold true, we obtain the probability that $\mathcal{B}$ cracks the CDH problem as $\epsilon' = \epsilon_1 \epsilon_2 \epsilon$. Since $\epsilon_1$, $\epsilon_2$, and $\epsilon$ are not negligible, $\epsilon'$ is not negligible. By the converse method, we conclude that PPT adversary $\mathcal{A}$ cannot crack our scheme with non-negligible probability $\epsilon$. The specific proof process is as follows.

**Theorem 1.** If there is a PPT adversary $\mathcal{A}$ who can break the proposed scheme with a non-negligible probability $\epsilon$, then there is another PPT algorithm $\mathcal{B}$ that can solve the CDH problem with a non-negligible probability $\epsilon' \geq e^2 \left(1 - \frac{1}{q_s q_h}\right)\left(1 - \frac{1}{q}\right)\epsilon$, where $q_s$ and $q_h$, respectively represent the number of Sign Query and $\boldsymbol{H_1}$ Query.

**Proof.** Suppose there is an adversary $\mathcal{A}$ that meets the above conditions, then we will construct another PPT algorithm $\mathcal{B}$, $\mathcal{B}$ will call $\mathcal{A}$ as a subroutine, and obtain $g^{ab}$ from the known public parameters $pp = (q, G_1, G_2, e, g)$ and $(g^a, g^b)$, where $g \in \mathbb{G}_1; a, b \leftarrow_R \mathbb{Z}_q^*$.

- **Setup:** $\mathcal{B}$ Selects a large integer $N$, then:
  1) Randomly selects $s_1, s_2, \ldots, s_N \in \mathbb{Z}_q^*$, calculates $g_j = \left(g^b\right)^{s_j}$ for $j \in [1, N]$;
  2) Lets $pk_A = g^a = u_A$, publishes the parameter $pp = (q, G_1, G_2, e, g, g_1, g_2, \ldots, g_N)$;
  3) Sends $pk_A$ and $pp$ to $\mathcal{A}$.

- **Combiner-Key Generation Query:** $\mathcal{A}$ will initiate multiple queries. $\mathcal{B}$ denotes the $t$-th query as $(pk_B^{(t)}, sk_B^{(t)})$, and guesses that the $T$-th query corresponds to the final forgery of $\mathcal{A}$. $\mathcal{B}$ creates a list $L_k$ to record this query, and each record in $L_k$ is $(t, pk_B^{(t)}, sk_B^{(t)})$. When $\mathcal{A}$ initiates this query, then $\mathcal{B}$:
  1) If $t \neq T$, $\mathcal{B}$ selects $y_t \leftarrow_R \mathbb{Z}_q^*$ as the private key $sk_B^{(t)}$, then calculates and returns the corresponding public key $pk_B^{(t)} = g^{y_t}$ to $\mathcal{A}$;
  2) If $t = T$, $\mathcal{B}$ selects $y_t \leftarrow_R \mathbb{Z}_q^*$, lets the public key $pk_B^{(t)} = g^{by_t} = u_B$. Neither $\mathcal{A}$ nor $\mathcal{B}$ knows the private key $sk_B^{(T)}$ corresponding to $pk_B^{(t)}$. $\mathcal{B}$ stores $\left(t, pk_B^{(t)}, sk_B^{(t)}\right)$ into list $L_k$ and returns $pk_B^{(t)}$ to $\mathcal{A}$.

- **Combiner Corruption Query:** When $\mathcal{A}$ initiates this query, $\mathcal{A}$ sends a combiner's public key $pk_B^{(t)}$ to $\mathcal{B}$. If $t = T$, $\mathcal{B}$ gives up the simulation, otherwise, $\mathcal{B}$ queries the list $L_k$ and returns $sk_B^{(t)}$ to $\mathcal{A}$.

- **$\boldsymbol{H_1}$ Query:** $\mathcal{B}$ builds a list $L_H$ to record the $\boldsymbol{H_1}$ Query, and each record in $L_H$ is $(\tau, \{\zeta_i, H_1(\tau, i)\}_{i=1}^m)$. When $\mathcal{A}$ initiates this query, $\mathcal{A}$ sends the subspace label $\tau$ to $\mathcal{B}$, then $\mathcal{B}$:
  1) If $\tau$ has already been queried, $\mathcal{B}$ queries the list $L_H$ and returns $\{H_1(\tau, i)\}_{i=1}^m$;
  2) Otherwise, $\mathcal{B}$ selects $\zeta_1, \zeta_2, \ldots, \zeta_m \leftarrow_R \mathbb{Z}_q^*$ and calculates $H_1(\tau, i) = \left(g^b\right)^{\zeta_i}$ for $i \in [1, m]$. $\mathcal{B}$ stores $\left(\tau, \{\zeta_i, H_1(\tau, i)\}_{i=1}^m\right)$ into list $L_H$ and returns $\{H_1(\tau, i)\}_{i=1}^m$.

- **Sign Query:** $\mathcal{A}$ queries for the signatures of the subspace $V \subset \mathbb{Z}_q^N$, then $\mathcal{B}$:
  1) Selects $id \leftarrow_R \{0, 1\}^\lambda$ and let the label of the vector subspace $V$ be $\tau = (id, pk_B^{(t)})$. If $H_1(\tau, \cdot)$ has already been queried, then $\mathcal{B}$ aborts the simulation;
  2) Lets $m = N - n$, calculates $\zeta_i = -\sum_{j=1}^n s_j v_{ij}$ for $i = 1, \ldots, m$;
  3) Selects a value $z_i \leftarrow_R \mathbb{Z}_q^*$, lets $H_1(\tau, i) = \frac{g^{z_i}\left(g^b\right)^{\zeta_i}}{pk_B^{(t)}}$;

4) Calculates $\sigma_i = g^{a(z_i + H_2(\mathbf{v}_i) - 1)}$;

5) Outputs label $\tau$ and signature $\{\sigma_i\}_{i=1}^m$.

- **Combine Query:** $\mathcal{A}$ sends $\left(pk_B^{(t)}, \tau, \{\beta_i, \mathbf{v}_i, \sigma_i\}_{i=1}^m\right)$ to $\mathcal{B}$, if $t = T$, then $\mathcal{B}$ gives up this simulation. Otherwise, then $\mathcal{B}$:

  1) Calculates $\mathbf{v} = \sum_{i=1}^m \beta_i \mathbf{v}_i$;

  2) Calculates $\sigma = g^{sk_B^{(t)} H_2(\mathbf{v}) a} \prod_{i=1}^m \left(\sigma_i \cdot (g^{sk_B^{(t)} H_2(\mathbf{v}_i) a})^{-1}\right)^{\beta_i}$;

  3) Outputs label $\tau$ and message/signature pair $(\mathbf{v}, \sigma)$.

- **Output:** In the above process, if $\mathcal{B}$ does not give up the simulation, the successful forgery of $\mathcal{A}$ means outputting a quadruple $(pk_B^*, \tau^*, \mathbf{v}^*, \sigma^*)$, where $\mathbf{v}^* \neq \mathbf{0}$, and **Verify** $(pp, pk_A, \tau^*, \mathbf{v}^*, \sigma^*) = 1$. If $\tau^*$ has not appeared in the signature query, $\mathcal{B}$ computes and outputs $g^{ab} = \left(\dfrac{\sigma^*}{g^{a(y + H_2(\mathbf{v}^*) - 1)}}\right)^{\frac{1}{\mathbf{s} \cdot \mathbf{v}^*}}$, where $\mathbf{s} = (s_1, \ldots, s_n, \zeta_1, \ldots, \zeta_m)$.

Below we prove that $\mathcal{B}$ successfully simulates the **Setup**, **KeyGen**, and **Sign** algorithm, and hash function $H_1$ without giving up the simulation. Since the **Combine** algorithm simulated by $\mathcal{B}$ runs completely according to the real algorithm, its correctness proof is ignored here.

Since $s_1, s_2, \ldots, s_N$ are randomly selected values, $g_1, g_2, \ldots, g_N$ are also random values, so $\mathcal{B}$ successfully simulates the algorithm **Setup** and **KeyGen**; and because $\zeta_1, \zeta_2, \ldots, \zeta_m$ are randomly selected values, the output of $H_1$ is also a random value, so $\mathcal{B}$ successfully simulates the hash function $H_1$. Below, we prove that $\mathcal{B}$ successfully simulates the **Sign** algorithm:

For the **Sign** algorithm, when the input parameter is $(pp, sk_A, pk_B^{(t)}, id, \mathbf{v})$, where $sk_A = a, pk_B^{(t)} = u_B$, the corresponding real signature value is:

$$\sigma = \left(\prod_{i=1}^n g_i^{v_i} \prod_{j=1}^m H_1(\tau, j)^{v_{n+j}} u_B^{H_2(\mathbf{v})}\right)^a. \tag{12}$$

Substituting the query value $g_i = (g^b)^{s_i}, H_1(\tau, j) = \dfrac{g^{z_j}(g^b)^{\zeta_j}}{u_B}, u_B = g^{y_t}$ into the above formula, the result of the **Sign Query** is:

$$
\begin{aligned}
\sigma &= \left(\prod_{i=1}^n (g^b)^{s_i v_i} \prod_{j=1}^m \left(\frac{g^{z_j}(g^b)^{\zeta_j}}{u_B}\right)^{v_{n+j}} g^{y_t H_2(\mathbf{v})}\right)^a \\
&= \left(g^{b\left(\sum_{i=1}^n s_i v_i + \sum_{j=1}^m \zeta_j v_{n+j}\right)} g^z g^{(H_2(\mathbf{v}) - 1)}\right)^a \\
&= g^{ab(\mathbf{s} \cdot \mathbf{v})} g^{a(z + H_2(\mathbf{v}) - 1)} \\
&= g^{a(z + H_2(\mathbf{v}) - 1)}.
\end{aligned}
\tag{13}
$$

According to the construction of $\zeta$ in the **Sign Query**, $\mathbf{s} \cdot \mathbf{v} = 0$ can be known, so the last equal sign in the above formula is established. It can be found that the output of the real

signature algorithm **Sign** is consistent with the output of $\mathcal{B}$, so $\mathcal{B}$ successfully simulates the algorithm **Sign**.

Below we analyze the probability that $\mathcal{B}$ does not give up the simulation. Let $q_k, q_r, q_h, q_s, q_c$ denote the query number of **Combiner-Key Generation Query**, **Combiner Corruption Query**, $H_1$ **Query**, **Sign Query**, and **Combine Query**, respectively. If $\mathcal{B}$ does not abandon the simulation, the following conditions need to be met during all queries:

1. The combiner public key corresponding to the final forged result of $\mathcal{A}$ was not used in $q_r$ times of **Combiner Corruption Query** initiated by $\mathcal{A}$, and this probability is $\left(1 - \frac{1}{q_k}\right)^{q_r}$;

2. In the $q_s$ **Sign Queries** initiated by $\mathcal{A}$, none of the vector subspace labels used by $\mathcal{B}$ has been queried by $\mathcal{A}$ in $H_1$ **Query** and this probability is $1 - \frac{1}{q_s \cdot q_h}$;

3. In the $q_c$ **Combine Queries** initiated by $\mathcal{A}$, the public key of the combiner corresponding to the final forged result of $\mathcal{A}$ is not used, and the probability is $\left(1 - \frac{1}{q_k}\right)^{q_c}$.

So the probability that $\mathcal{B}$ does not abandon the simulation is $\left(1 - \frac{1}{q_k}\right)^{q_r}\left(1 - \frac{1}{q_s \cdot q_h}\right)\left(1 - \frac{1}{q_k}\right)^{q_c} \geq e^2\left(1 - \frac{1}{q_s \cdot q_h}\right)$. Below we prove the correctness of the output of $\mathcal{B}$ when $\mathcal{B}$ does not give up on the simulation. Let $\mathcal{A}$ finally outputs the quadruple $(pk_B^*, \tau^*, \mathbf{v}^*, \sigma^*)$, and **Verify**$(pp, pk_A, \tau^*, \mathbf{v}^*, \sigma^*) = 1$, then

$$e(\sigma^*, g) = e\left(\prod_{i=1}^{n} g_i^{v_i^*} \prod_{j=1}^{m} H_1(\tau, j)^{v_{n+j}^*} u_B^{H_2(\mathbf{v}^*)}, u_A\right). \tag{14}$$

Substituting the query value $g_i = (g^b)^{s_i}, H_1(\tau^*, j) = (g^b)^{\zeta_j}, u_B = g^y$ into the above formula, we get

$$e(\sigma^*, g) = e\left(g^{b\left(\sum_{i=1}^{n} s_i v_i^* + \sum_{j=1}^{m} \zeta_j v_{n+j}^*\right)} g^y g^{H_2(\mathbf{v}^*)-1}, u_A\right)$$
$$= e\left(g^{ab(\mathbf{s} \cdot \mathbf{v}^*)} g^{a(y + H_2(\mathbf{v}^*)-1)}, g\right). \tag{15}$$

Therefore $\sigma^* = g^{ab(\mathbf{s} \cdot \mathbf{v}^*)} g^{a(y + H_2(\mathbf{v}^*)-1)}$, if $\mathbf{s} \cdot \mathbf{v}^* \neq 0$, then $g^{ab} = \left(\frac{\sigma^*}{g^{a(y+H_2(\mathbf{v}^*)-1)}}\right)^{\frac{1}{\mathbf{s} \cdot \mathbf{v}^*}}$. If $\mathbf{s} \cdot \mathbf{v}^* = 0$, $\mathcal{B}$ will not output the value of $g^{ab}$ correctly. Event $\mathbf{s} \cdot \mathbf{v}^* = 0$ occurs in the following three situations:

1. When the forgery of $\mathcal{A}$ belongs to the Type 1 Forgery. Since all values of $\mathbf{s}$ are randomly selected numbers in the space $\mathbb{Z}_q$, and $\mathbf{v}^* \neq \mathbf{0}$, so $\mathbf{s} \cdot \mathbf{v}^*$ is uniformly distributed in $\mathbb{Z}_q$, then $P(\mathbf{s} \cdot \mathbf{v}^* = 0) = \frac{1}{q}$.

2. When the forgery of $\mathcal{A}$ belongs to the Type 2 Forgery. Since all values of $\mathbf{s}$ are randomly selected numbers in the space $\mathbb{Z}_q$, then $P(\mathbf{s} \cdot \mathbf{v}^* = 0) = \frac{1}{q}$ in the same way.

3. When the forgery of $\mathcal{A}$ belongs to the Type 3 Forgery. All values of $\mathbf{s}$ are randomly selected numbers in space $\mathbb{Z}_q$, $\mathbf{v}^* \in V - \{\mathbf{v}_1, \mathbf{v}_2, \ldots, \mathbf{v}_m\}$ and $\mathbf{v}^* \neq \mathbf{0}$, so the value of $\mathbf{s} \cdot \mathbf{v}^*$ is uniformly distributed in $\mathbb{Z}_{q-m}$, then $P(\mathbf{s} \cdot \mathbf{v}^* = 0) = \frac{1}{q-m}$. Because of $q \gg m$, $P(\mathbf{s} \cdot \mathbf{v}^* = 0)$ at this time is close to $\frac{1}{q}$.

In summary, the probability of event $\mathbf{s} \cdot \mathbf{v}^* \neq \mathbf{0}$ is $1 - \frac{1}{q}$. We set the probability that $\mathcal{A}$ successfully outputs a valid signature as $\epsilon$, then $\mathcal{B}$ can correctly output the value of $g^{ab}$ with probability $\epsilon' \geq e^2 \left(1 - \frac{1}{q_s \cdot q_h}\right)\left(1 - \frac{1}{q}\right)\epsilon$.

Since the CDH assumption is established, the probability $\epsilon'$ of $\mathcal{B}$ correctly outputting $g^{ab}$ is negligible, so the probability $\epsilon$ is negligible.

## APPLICATION AND EXPERIMENT ANALYSIS

Here, we illustrate how the proposed scheme works when it is applied to proxy signatures, and then theoretically analyzed the efficiency of our scheme. Finally, the signature forgery experiments on LZZ22 and our scheme are respectively run in the same experimental environment, and the efficiency of our scheme is compared with other schemes.

### Application

Digital signature technology can provide authenticity and integrity certification to users. In real life, a large number of signature activities are often required in some departments (*e.g.*, governments and hospitals). Ordinary digital signature schemes do not allow entities other than a specific user to have signing privileges, so users have to accomplish a large number of signing tasks on their own. The linearly homomorphic signature scheme with a designated combiner can improve the efficiency of signing by transferring the user's large number of computational tasks to a server with high computational power. Consider the following specific scenario:

Suppose the user has partitioned the file set labeled $\tau$ into $m$ $n$-dimensional vectors $\overline{\mathbf{v}}_1, \overline{\mathbf{v}}_2, \ldots, \overline{\mathbf{v}}_m \in \mathbb{Z}_q^n$, and augmented $\{\overline{\mathbf{v}}_i\}_{i=1}^m$ into a set of basis vectors $\{\mathbf{v}_i\}_{i=1}^m$ in the subspace $V$ according to the method in "Preliminaries". The user has computed the signatures $\{\sigma_k\}_{k=1}^m$ of $\{\mathbf{v}_i\}_{i=1}^m$ respectively. At this time, if the user needs to generate the signatures $\sigma^*$ of the data vectors $\mathbf{v}^* = \sum_{k=1}^m \beta_k \mathbf{v}_k$, $\beta_k \in \mathbb{Z}_q^*$, the user can only re-compute the signature on his own in the ordinary digital signature scheme. In contrast, in the LHSDC scheme, the user only needs to send the label $\tau$ and the combination coefficients $\beta_k$ to the specified server, and the specified server will complete the signature instead of the user (as shown in Fig. 1). We call this application scenario proxy signing.

The proposed scheme contains three types of participants when applied to proxy signing, namely the signer (user), the designated combiner (server) and the verifier (Fig. 2). The specific application process is as follows:

**Step 1:** The system runs the algorithm **Setup** to generate the system public parameter $pp$ and publish $pp$ to all participants. The signer and the designated combiner run the algorithm **KeyGen** respectively to generate the private/public key pair $(sk_A, pk_A)$ of the signer and the private/public key pair $(sk_B, pk_B)$ of the designated combiner.

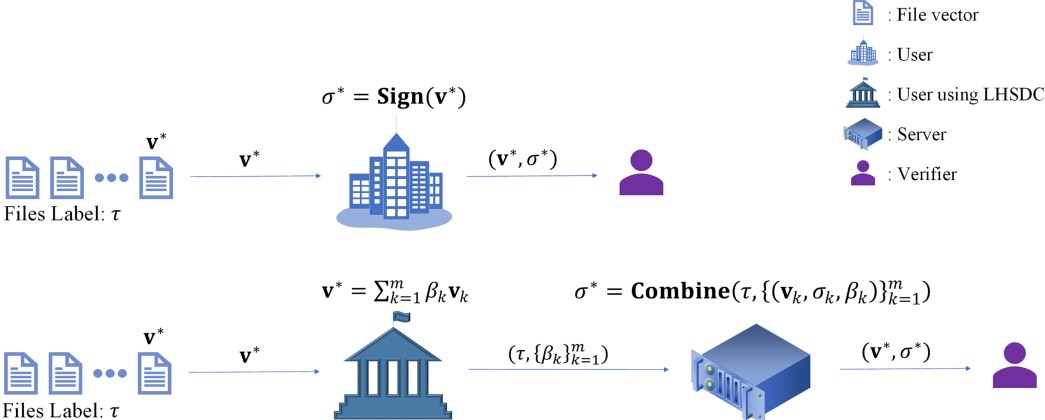

**Figure 1** **The advantage of LHSDC in proxy signing.**

**Step 2:** The signer first divides the message file needing a signature into $m$ $n$-dimensional message vectors $\overline{\mathbf{v}}_1, \overline{\mathbf{v}}_2, \ldots, \overline{\mathbf{v}}_m \in \mathbb{Z}_q^n$, and uses the vector augmentation method in "Preliminaries" section to augment each $n$-dimensional message vector into an $N$-dimensional subspace basis vector $\mathbf{v}_1, \mathbf{v}_2, \ldots, \mathbf{v}_m \in \mathbb{Z}_q^N$. Then, the label $\tau$ of the vector subspace is generated according to the file identifier *id*. Finally, the signer runs the algorithm **Sign** to sign $\{\mathbf{v}_k\}_{k=1}^m$ to obtain $\{\sigma_k\}_{k=1}^m$, send $(\tau, \{(\mathbf{v}_k, \sigma_k)\}_{k=1}^m)$ to the designated combiner, which runs the algorithm **Verify** to verify the legitimacy of $\{\sigma_k\}_{k=1}^m$ respectively. If $\{\sigma_k\}_{k=1}^m$ are all valid, the designated combiner will store them.

**Step 3:** When the signer wants to generate the signature $\sigma^*$ of the new message $\mathbf{v}^*$ under this subspace, he only needs to express $\mathbf{v}^*$ as $\mathbf{v}^* = \sum_{k=1}^m \beta_k \mathbf{v}_k$, and then send the combination coefficient $\{\beta_k\}_{k=1}^m$ to the designated combiner. The designated combiner runs the algorithm **Combine** to get $(\mathbf{v}^*, \sigma^*)$, and sends $(\mathbf{v}^*, \sigma^*)$ to the verifier. The verifier runs the algorithm **Verify** to verify the legitimacy of the combined signature $\sigma^*$.

In proxy signing, let us consider another case. After the designated combiner has derived the signature $\sigma^*$ of a certain message $\mathbf{v}^*$ by the algorithm **Combine**, the signer expects the combiner to continue generating the signature $\sigma^{**}$ of the message $\mathbf{v}^{**} = n\mathbf{v}^*$. At this point, $(\mathbf{v}^*, \sigma^*)$ cannot be used as an input to the algorithm **Combine** because the signature $\sigma^*$ generated by the combiner is structurally altered compared to the original signature generated by the signer in the LHSDC. The signer must first decompose $\mathbf{v}^{**}$ into $\sum_{k=1}^m \beta_k^* \mathbf{v}_k, \ \beta_k^* \in \mathbb{Z}_q^*$, and then the combiner inputs $\{\mathbf{v}_k, \sigma_k, \beta_k^*\}_{k=1}^m$ into the algorithm Combine to obtain the signature $\sigma^{**}$. *Boneh et al. (2009)* determined that the process requires $(m+1)$ pairing operations, $(m+2)$ exponentiation operations, $m$ inverse operations, $2m$ multiplication operations, and $(2m+1)$ hash operations. In contrast, SPS-LHSDC maintains the signature structure on top of the function of designating a combiner. Therefore, $\mathbf{v}^{**}$ does not need to be decomposed into the form of multiple basis vector representations, and $(\mathbf{v}^*, \sigma^*, n)$ can be directly used as the input to the algorithm Combine. Using our scheme, $\sigma^{**} = u_A^{a_B H_2(\mathbf{v}^{**})} \left( \sigma^* \left( u_A^{a_B H_2(\mathbf{v}^*)} \right)^{-1} \right)^n$, the process requires

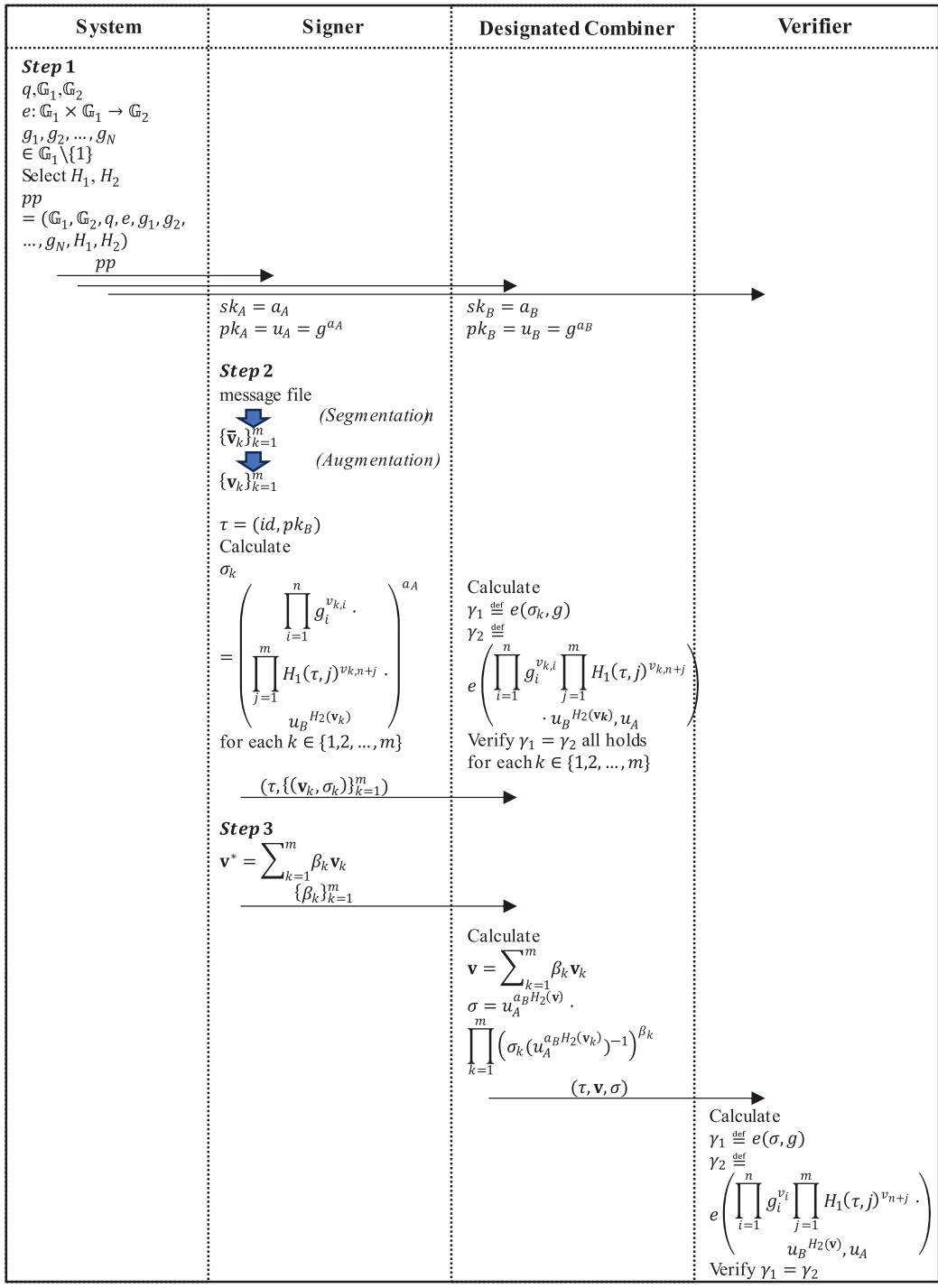

**Figure 2 The application process of the proposed scheme.**

only three exponential operations, one inverse operation, four multiplication operations, and two hash operations. The SPS-LHSDC scheme can significantly reduce the computation of proxy signing as shown in Fig. 3.

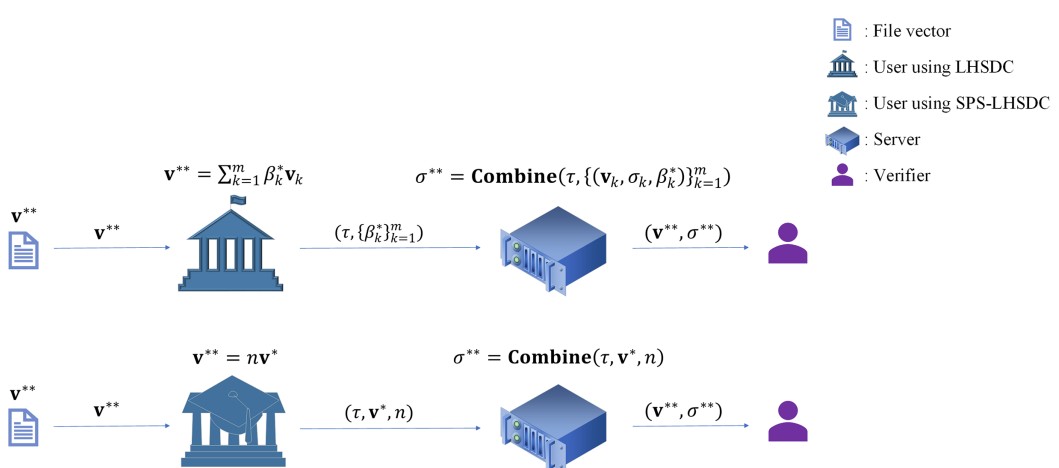

**Figure 3  The advantage of SPS-LHSDC in proxy signing.**

**Table 1  Notations and the correspondent operations.**

| Notation | Operation |
|---|---|
| $H$ | Map-to-point hash operation |
| $E_1$ | Exponential operation in $\mathbb{G}_1$ |
| $E_2$ | Exponential operation in $\mathbb{G}_2$ |
| $M_1$ | Multiplicative operation in $\mathbb{G}_1$ |
| $M_2$ | Multiplicative operation in $\mathbb{G}_2$ |
| $P$ | Bilinear pairing operation |
| $|\mathbb{G}_1|$ | The size of elements in $\mathbb{G}_1$ |
| $|\mathbb{G}_2|$ | The size of elements in $\mathbb{G}_2$, $|\mathbb{G}_2| < |\mathbb{G}_1|$ |

In summary, SPS-LHSDC can reduce the calculation of the signer and improve the efficiency of signing compared with ordinary digital signature schemes and LHSDC schemes. However, since the designated combiner (server) in SPS-LHSDC also has the authority to generate signatures, the server must be under the management of the most authoritative department of the organization. Meanwhile, the SPS-LHSDC scheme does not apply to some departments with high confidentiality due to the irreplaceable nature of the signatures of these departments.

## Theoretical analysis

Table 1 illustrates the meanings of the notations used in this section. Table 2 compares our scheme with the other four schemes in *Lin, Xue & Huang (2021)*, *Li, Zhang & Zhang (2022)*, *Zhang et al. (2018)* and *Wu, Wang & Yao (2021)* in terms of efficiency and functionality.

By comparing with other four schemes theoretically, we find that only our scheme and LZZ22 are able to realize both functions of designating a combiner and maintaining the signature structure. In "The Security Problem of LZZ22", the LZZ22 scheme was shown to have a security vulnerability, and our scheme was the only secure SPS-LHSDC scheme.

**Table 2 Comparison of cost and functions.**

| Scheme | Sig.cost | Verify.cost | Signature size | Characteristic |
|---|---|---|---|---|
| Lin21 | $3H + (n+2)E_2 + (n+1)M_2$ | $(m+2)H + (n+1)E_2 + nM_2 + 3P$ | $|\mathbb{G}_2|$ | LHSDC |
| LZZ22 | $H + (n+1)E_1 + (n+1)M_1$ | $mH + nE_1 + (n+1)M_1 + 2P$ | $|\mathbb{G}_1|$ | SPS-LHSDC |
| Zhang18 | $(m+n)H + (n+2)E_2 + (n-1)M_2$ | $(m+2)H + (n+1)E_2 + nM_2 + 2P$ | $|\mathbb{G}_2|$ | ID-based |
| Wu21 | $(n+1)H + (n+2)E_2 + (n+1)M_2$ | $(n+1)H + (n+1)E_2 + (n-1)M_2 + 4P$ | $2|\mathbb{G}_1|$ | Certificateless |
| Our | $2H + (n+2)E_1 + (n+1)M_1$ | $(m+1)H + (n+1)E_1 + (n+1)M_1 + 2P$ | $|\mathbb{G}_1|$ | SPS-LHSDC |

Our scheme has the lowest computational overhead for both the signature algorithm and the verification algorithm, thus our scheme is efficient.

## Experiment analysis

In this section, we run the signature forgery program of LZZ22 through experiments, so as to obtain the probability and time required for an adversary to successfully forge a signature. Then, under the same experimental environment, we run our scheme and evaluate its efficiency.

The following illustrates experimental environment and parameter selection. We build the simulator in Python and use a 2.6 GHz single-core twelve-thread processor. The parameter params we used in LZZ22's, Wu21's and our simulations are from pypbc (*Maas, 2004*) library's A-type curve. The parameter params we used in Lin's and Zhang18's simulations are from pypbc library's F-type curve. The security parameter length is 80 bits, and the element lengths in $\mathbb{G}_1$ and $\mathbb{G}_2$ are 320 and 160 bits respectively. In our experiment, in order to meet the needs of simulating multiple scenarios, the size of the test file we choose is 3.2 KB (3,279 bytes). The file will be divided into $m$ blocks, each block contains $n$ elements, and each element length is 160 bits, which means that the test file is represented by $m$ $n$-dimensional vectors, and the values of $m$ and $n$ need to meet: $\frac{160}{8}(m-1)n \le 3,279 \le \frac{160}{8}mn$. According to the load of the network, we set the packet size as 1,460 bytes. Thus, each packet can hold $\frac{1,460}{20} = 73$ elements. Therefore, the augmented data packet length $N$ should satisfy: $N = m + n \le 73$.

### Signature forgery experiment of LZZ22

The signature forgery of LZZ22 includes two processes, namely the signature query and the signature forgery. Since the adversary can obtain the signatures $\sigma_0, \sigma_{00}$, and $\{\sigma_i\}_{i=1}^m$ from the signatures generated by the signer in the past, we ignore the cost of the signature query. The cost of the signature forgery process is consistent with the formula

$\sigma^* = \left( \prod_{i=1}^m \left( \frac{\sigma_i \sigma_{00}}{\sigma_0^2} \right)^{\beta_i} \right) \cdot \frac{\sigma_0^2}{\sigma_{00}}$. Figure 4 shows the relationship between the cost required for

the signature forgery and the message vector dimension $n$.

From Fig. 4, it can be found that the adversary can forge the signature of LZZ22 in a very small amount of time after the **Sign Query**. The time required for forging the signature decreases with the increase of the dimension $n$. This is because an increase in $n$ is accompanied by a decrease in $m$.

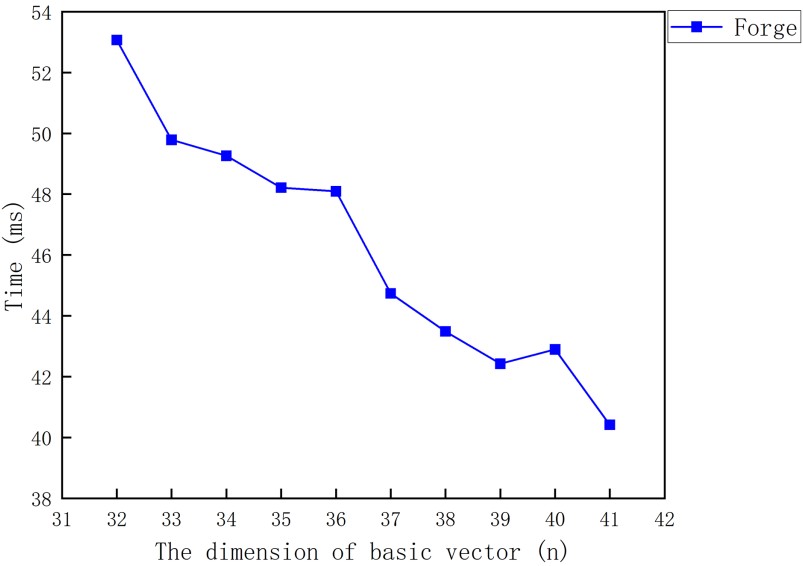

**Figure 4 The cost of the signature forgery for LZZ22.**

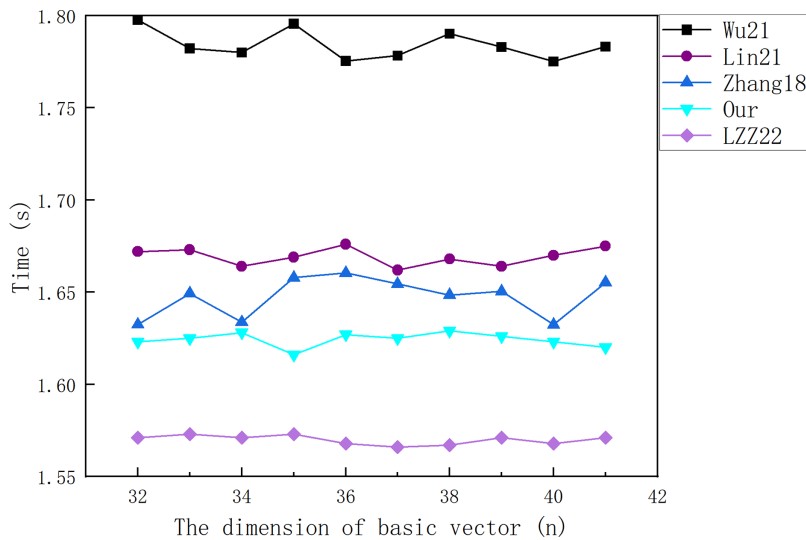

**Figure 5 The cost of signing a 3.2 KB file.**

### Efficiency analysis experiment of the proposed scheme

Figures 5 and 6 show a comparison between our scheme and the schemes in *Lin, Xue & Huang (2021)*, *Li, Zhang & Zhang (2022)*, *Zhang et al. (2018)* and *Wu, Wang & Yao (2021)* from the cost of the signature algorithm and verification algorithm, respectively. Figure 7 shows the CPU and RAM occupancy of each scheme, and Fig. 8 shows the specific usage of RAM for each scheme.

Figure 5 illustrates that the cost of each scheme in the signature algorithm does not change greatly with the increase of the dimension value $n$ of the base vector. This is because an increase in $n$ is accompanied by a decrease in $m$, that is, although the length of each data packet increases, a file can be represented using fewer data packets. Figure 6 shows that the

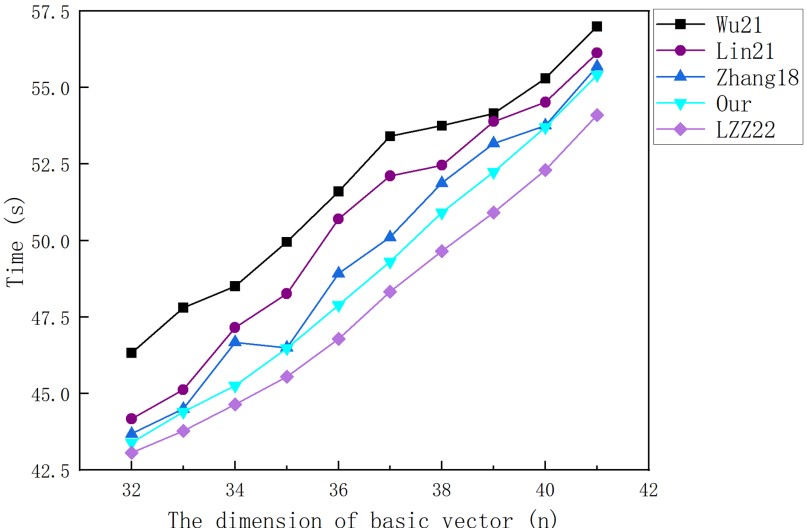

**Figure 6 The cost of verifying a message vector.**

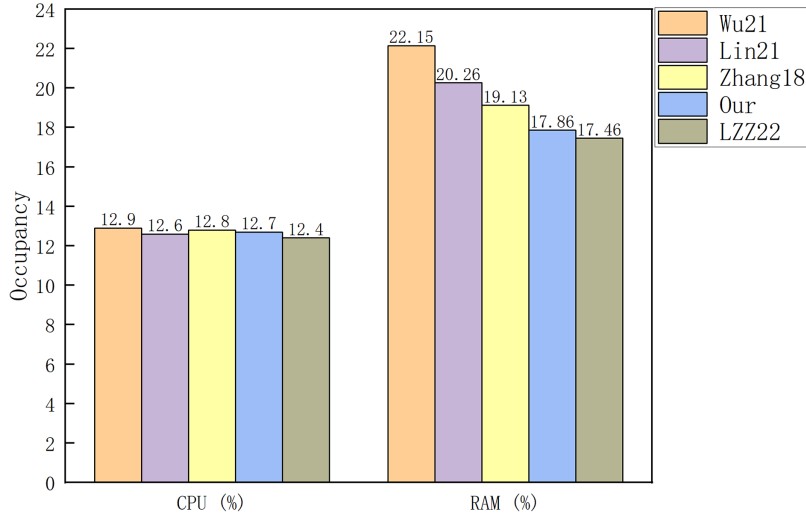

**Figure 7 Occupancy of CPU and RAM.**

cost of each scheme in the verification algorithm increases with the increase of the dimension $n$ of the base vector. This is because the verification of each vector needs to add one multiplication operation and one exponent operation due to the increase of $n$.

By comparing our scheme with the experimental results of other schemes, we find that whether signing a 3.2 KB file or verifying a single message vector, the time overhead of the LZZ22 scheme is the smallest, and our scheme is the second smallest. However, we have proved in "The Security Problem of LZZ22" that the LZZ22 scheme has a security vulnerability. Therefore, our scheme has the highest efficiency among the remaining schemes. When the amount of valid data in the packet reaches the maximum value of 41 ($n = 41$), our scheme takes 1.620 s to sign a 3.2 KB file and 55.406 ms to verify a single data vector under that file.

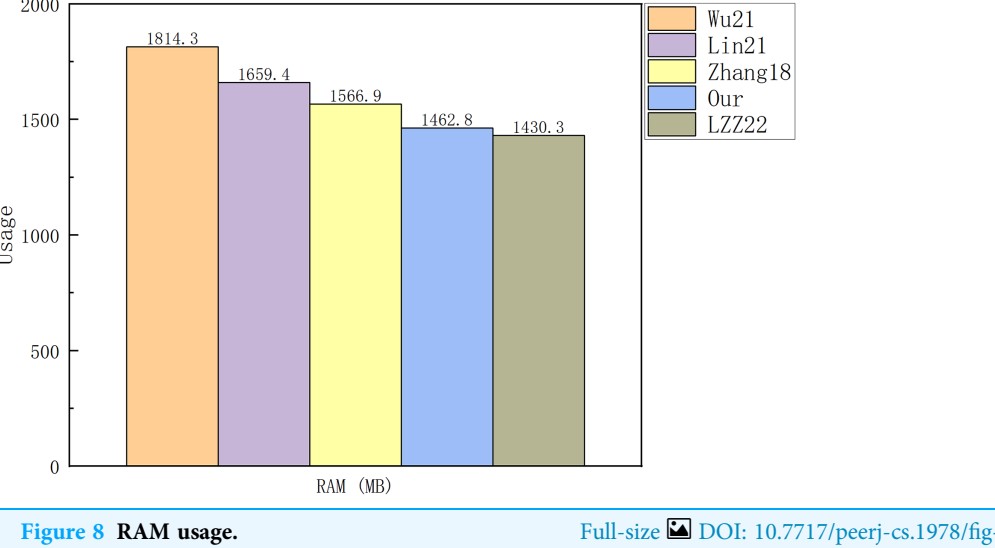

**Figure 8** **RAM usage.**

Figures 7 and 8 illustrate that there is no major difference between the schemes in terms of CPU occupancy. In terms of RAM occupancy, LZZ22 has the smallest memory usage, followed by our scheme. Since the LZZ22 scheme is insecure, our scheme has the smallest usage on RAM among the remaining schemes. When the file size is 3.2 KB, running our scheme will consume 12.7% of CPU and $\frac{1,462.8}{8*1,024} = 17.86\%$ of system memory. Overall, our scheme has a low computational overhead and system resource usage on top of the simultaneous functionality of designating a combiner and maintaining the signature structure.

## CONCLUSION

Here, we prove that there is a polynomial time adversary that can crack the secret information in LZZ22 through multiple signature queries, which will allow the adversary to forge the signature corresponding to any message. We proposed a new scheme, which has all the functions of LZZ22 and fixed the security problem by adding one hash operation and one exponential operation to the signature algorithm. Our scheme proved secure against existential forgery on adaptively chosen subspace attacks under the random oracle model. We also detailed the application of our scheme to the proxy signature. Finally, we ran the signature forgery program of LZZ22 through experiments, and the results showed that the time required to forge a signature was inversely proportional to the message dimension. The proposed scheme was run in the same experimental environment and compared with other similar schemes. The experimental results show that the signature algorithm and the verification algorithm of our scheme are efficient, and efficiently use the system resources.

It should be noted that in proxy signing because our scheme allows the designated combiner (server) to generate legal signatures, the server must be managed at the highest level in the department. Meanwhile, the SPS-LHSDC scheme does not apply to some departments with high confidentiality due to the irreplaceable nature of the signatures of these departments. In addition, in this article, we only explore the application of SPS-

LHSDC scheme in proxy signatures, and some other application scenarios that need to specify servers for calculating such as federated learning, cloud auditing, and so on deserve more in-depth research. Although the proposed scheme is more efficient than the existing LHSDC scheme and is the only secure SPS-LHSDC scheme, there is still room for improving the efficiency of the signature algorithm. Two directions deserve further research to improve the efficiency of the signature algorithm. One of them is to optimize the homomorphic hash function, which can significantly improve the efficiency of the scheme. The other is to optimize the way to bind combiners, that is, to find a way to bind combiners other than key exchange, which can improve the efficiency of the scheme by a small margin.

### Funding
This work was supported by the National Natural Science Foundation of China (Grant Nos. 62172436, 62102452), the National Key R&D Program of China (2021YFB3100100), Innovative Research Team in Engineering University of PAP (KYTD201805), and the Natural Science Foundation of Shaanxi Province (2023-JC-YB-584). The funders had no role in study design, data collection and analysis, decision to publish, or preparation of the manuscript.

### Grant Disclosures
The following grant information was disclosed by the authors:
National Natural Science Foundation of China: 62172436, 62102452.
National Key R&D Program of China: 2021YFB3100100.
Innovative Research Team in Engineering University of PAP: KYTD201805.
Natural Science Foundation of Shaanxi Province: 2023-JC-YB-584.

### Competing Interests
The authors declare that they have no competing interests.

### Author Contributions
- Xuan Zhou conceived and designed the experiments, performed the experiments, performed the computation work, prepared figures and/or tables, and approved the final draft.
- Yuan Tian performed the experiments, analyzed the data, performed the computation work, prepared figures and/or tables, and approved the final draft.
- Weidong Zhong analyzed the data, authored or reviewed drafts of the article, and approved the final draft.
- Tanping Zhou analyzed the data, authored or reviewed drafts of the article, and approved the final draft.
- Xiaoyuan Yang analyzed the data, authored or reviewed drafts of the article, and approved the final draft.

## Data Availability

The raw data and code are available in the Supplemental Files.

## Supplemental Information

Supplemental information for this article can be found online at http://dx.doi.org/10.7717/peerj-cs.1978#supplemental-information.

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
