# Peer review of "A structure-preserving linearly homomorphic signature scheme with designated combiner"

_PeerJ Computer Science, doi:10.7717/peerj-cs.1978_

## Round 0.1 · original submission · Major Revisions

Based on my opinion, I think the paper should be improved according to the reviewers' suggestions.

**Language Note:** The review process has identified that the English language must be improved. PeerJ can provide language editing services - please contact us at copyediting@peerj.com for pricing (be sure to provide your manuscript number and title). Alternatively, you should make your own arrangements to improve the language quality and provide details in your response letter. – PeerJ Staff

Reviewer 1 ·

Basic reporting

The manuscript adeptly presents a review of the initial SPS-LHSDC scheme LZZ22 proposed by Li et al. However, upon scrutinizing its security, the authors discern a critical flaw—an identified vulnerability allowing a polynomial time adversary to compromise the secret in LZZ22 through multiple signature queries. Subsequently, the manuscript introduces an enhanced scheme aimed at rectifying this security loophole by implementing a dynamic change in the constant secret concerning the message. Furthermore, a security analysis of the improved scheme is furnished under the random oracle model. Notably, the improved scheme's application to the proxy signature is explored, culminating in a performance evaluation. However, it is essential to acknowledge that the authors’ work exhibits certain shortcomings, which I will elucidate in greater detail below. In conclusion, I am of the opinion that the current iteration of the manuscript falls short of the requisite quality for publication:

1) The introduction section lacks a clear outline of the paper's structure and content. Providing an outline at the outset would enhance the clarity and coherence of the manuscript, aiding readers in understanding the subsequent sections and their relevance to the overall research focus.

2) The paper attempts to address deficiencies in the LZZ22 method previously published. However, its contribution is somewhat constrained and exhibits certain limitations. The new vulnerability introduced in the proposed scheme appears to be specific and not widely observed in other related works, raising concerns about its applicability to broader algorithms. Consequently, considering these limitations, the paper's overall utility is deemed insufficient. Hence, the significance of highlighting this point for wider algorithmic applications appears questionable.

Experimental design

The authors are encouraged to bolster their experimental section by incorporating a more comprehensive set of comparative literature. Presently, the literature review seems limited, as it only includes two references for comparison, which may not adequately emphasize the advantages of the protocol. Expanding the range of referenced literature will enhance the credibility and persuasiveness of the experimental findings.

Validity of the findings

The Conclusion section lacks sufficient exploration of the current limitations inherent in the proposed scheme. It fails to adequately address the potential shortcomings or constraints that may affect the effectiveness or practicality of the solution presented in the paper. Furthermore, the future work outlined in the Conclusion appears vague and lacking in specificity. It would greatly benefit from a more detailed delineation of the avenues for future research, including specific areas for improvement, further experimentation, or potential extensions to the proposed scheme. Enhancing the Conclusion section with a comprehensive analysis of limitations and a clearer roadmap for future work would significantly strengthen the paper.

Additional comments

1) The present version exhibits several grammatical errors that require attention. I recommend the authors to revise the paper with the assistance of a fluent English speaker. For instance, within the Abstract section, the sentence, “Then, we propose a new scheme, which realize all the functions of LZZ22 and fix the security problem by making the above secret change with the message,” needs correction. Specifically, “realize” and “fix” should be substituted with “realizes” and “fixes”, respectively.
2) The authors need to provide a clearer definition and explanation of what constitutes a pollution attack within the context of the article for better comprehension.
3) The authors have meticulously detailed the PRELIMINARIES section; however, it lacks proper referencing, leading to a concern regarding the absence of citations for the information presented. The content in this section appears to be original, without attribution to external sources. Moving forward, it’s crucial to ensure that appropriate references are included to acknowledge the origins of the information presented, aligning with academic standards and giving due credit to prior work in the field.
4) The manuscript presents a significant issue with the formatting of references. Several references, such as the example provided (“Li, Y., Zhang, F., and Liu, X. (2022a). Secure data delivery with identity-based linearly homo morphic network coding signature scheme in IoT. IEEE Transactions on Services Computing, 15(4):2202–2212. Conference Name: IEEE Transactions on Services Computing.” and “Li, T., Chen,W., Tang, Y., and Yan, H. (2018). A homomorphic network coding signature scheme for multiple sources and its application in IoT. Security and communication networks, 2018. Publisher: Hindawi.”), are incorrectly formatted. The authors are urged to rectify these formatting errors to adhere to the standard reference format prescribed by PeerJ Computer Science. Accurate and consistent referencing is fundamental in academic writing, and rectifying these errors is imperative to ensure the manuscript meets the journal’s formatting guidelines and maintains academic integrity.
5) In the manuscript, “Computational Diffie-Hellman Problem (CDH)” and “Linearly homomorphic signature (LHS)”, there exists inconsistency in the way abbreviations are defined. The authors need to adopt a consistent approach by either capitalizing the first letter of each word in the abbreviation or maintaining all initial words in lowercase. This will ensure uniformity and clarity throughout the manuscript.

Reviewer 2 ·

Basic reporting

Introduction Clarity:

The introduction could be more explicit in outlining the significance of the public composability issue in LHS and how it hinders specific scenarios. Clarifying these points will help readers understand the motivation behind seeking improvements.
LZZ22 Evaluation:

Provide a more in-depth evaluation of the LZZ22 scheme's security features before introducing the identified flaw. This will help readers appreciate the context and significance of the proposed improvements.

Security Analysis:
The paper asserts the security vulnerability in LZZ22 but lacks a detailed analysis. Providing a step-by-step breakdown of the identified weakness and the polynomial time adversary's methodology will enhance the paper's academic rigor.

Proposed Scheme Presentation:
While introducing the proposed scheme, provide a clearer comparison with LZZ22, emphasizing the specific modifications that address the security flaw. A detailed exposition of the scheme's components and rationale will aid comprehension.


Empirical Results:
The section detailing the application of the proposed scheme on a personal computer lacks specificity. Include details such as computational complexity, time efficiency, and resource utilization to quantify the scheme's practical feasibility.

Experimental design

Proxy Signature Application:
Expand on the section detailing the application of the scheme to proxy signatures. Provide use-case scenarios, potential advantages, and any limitations or challenges encountered during implementation.

Validity of the findings

Theoretical Proof:
Elaborate on the theoretical proof of the proposed scheme's security. Provide a comprehensive walkthrough of the proof steps, assumptions made, and the logical flow to bolster the readers' confidence in the solution's robustness.

---

## Round 0.2 · accepted · Accept

The paper has addressed all reviewers' questions.

Reviewer 1 ·

Basic reporting

The authors have done all the corrections given in the previous round. Therefore, the paper can be accepted in the present format.

Experimental design

None

Validity of the findings

None

Additional comments

None

Reviewer 2 ·

Basic reporting

Authors updated the paper and no more update needed from my side.

Experimental design

As above

Validity of the findings

As above